# Girdling increases branch capacity to rehydrate in *Juniperus thurifera* and drought hampers bimodal growth

J. Julio Camarero[1] , Roberto L. Salomon[2], Antonio Gazol[1], Cristina Valeriano[1], Elisa Tamudo[1], Alvaro Rubio-Cuadrado[3] and Michele Colangelo[4]

[1]Pyrenean Institute of Ecology (IPE-CSIC), Spain; [2]Polytechnic University of Madrid, Spain; [3]Instituto Nacional de Investigación y Tecnología Agraria y Alimentaria, Spain; [4]Universita degli Studi della Basilicata, Italy

**Original Research Article**

**Keywords:**
drought; intra-annual wood density fluctuation; *Juniperus thurifera*; non-structural carbohydrates; point dendrometer.

**Corresponding author:**
J. Julio Camarero;
Email: jjcamarero@ipe.csic.es

**Associate Editor:**
Dr. Félix Hartmann

## Abstract

Disentangling how forests respond to aridification in terms of carbon storage and use, including bimodal growth, is critical to forecast their mitigation potential. Bimodality, characteristic of Mediterranean trees, refers to the potential to produce a second growth peak after the dry summer, often accompanied by intra-annual wood density fluctuations (IADF). To induce IADF formation, we performed a girdling experiment on Spanish juniper (*Juniperus thurifera*) branches in a semi-arid site, and monitored changes in branch diameter, and measured non-structural carbohydrate (NSC) concentrations in sapwood and leaves. IADFs were formed in response to wet conditions in late summer in girdled and non-girdled branches. After girdling, the extraordinarily dry 2022 growing season hampered branch radial increment and IADF production. Girdled branches swelled more than control branches after rain pulses. This suggests girdled branches were osmotically more active. Girdled branches presented higher starch leaf concentrations, suggesting that osmolytes could proceed from starch hydrolysis upstream. Girdling did neither trigger bimodal growth nor IADF formation during a very dry year.

## 1. Introduction

The cambium is a major meristem in terrestrial ecosystems (Larson, 1994), leading to the formation of wood, which constitutes one of the main terrestrial carbon pools (Cuny et al., 2015). Climate warming is intensifying aridification, thus reducing tree growth worldwide (Babst et al., 2019), which limits the potential to mitigate $CO_2$ emissions of seasonally dry forests. This is the case of Mediterranean forests, where trees respond to drought through bimodal radial growth with peaks during warm-wet seasons, i.e. spring and autumn (Camarero et al., 2010; Cherubini et al., 2003). However, we lack a mechanistic understanding of how bimodal growth responds to carbon availability and drought stress, which is required to assess long-term changes of carbon uptake in seasonally dry forest ecosystems.

Bimodal growth patterns reflect the sensitivity of radial growth to seasonal climate conditions and have been mainly reported for conifers (but see Camarero et al., 2024), including isohydric ('water saver') pines (e.g., *Pinus halepensis* Mill., *Pinus pinaster* Ait.) and also anisohydric ('water spender') junipers (e.g., *Juniperus phoenicea* L., *Juniperus thurifera* L.) (Liphschitz et al., 1984; Valeriano et al., 2023). Bimodality may be a facultative phenomenon driven by site climate conditions (e.g., aridity, continentality), species-specific xylogenesis or year-to-year shifts in spring and autumn precipitation (Campelo et al., 2021; Pacheco et al., 2016, 2018; Tumajer et al., 2021; Valeriano et al., 2023). In many of these studies, bimodality is related to the formation of latewood intra-annual wood density fluctuations (IADFs), characterized by earlywood-like tracheids with wide lumens and thin cell walls formed within the latewood (Camarero et al., 2023; De Micco et al., 2016; Olano et al., 2015; Vieira et al., 2009).

Bimodal growth and latewood IADFs are triggered by wet-cool conditions in late summer and early autumn, resuming cambial activity (carbon sink) (Collado et al., 2018; Rozas et al., 2011), or by manipulating carbon allocation through girdling (Oberhuber et al., 2017; 2021). Because carbon sinks and sources are modulated by different cues, bimodality and IADF formation are not necessarily linked (Rigling et al., 2002; Wimmer et al., 2000). For instance, Oberhuber et al. (2021) induced bimodality in a mountain Scots pine (*Pinus sylvestris* L.) stand, usually showing a spring growth peak (unimodal growth), by blocking phloem transport, thus shifting carbon availability and leading to the resumption of cambial activity above the girdle after about 2 weeks of treatment application. Thus, facultative bimodality offers an interesting setting to assess the relative roles played by carbon sinks and sources on radial growth in seasonally dry areas by manipulating carbon availability through girdling.

Girdling is widely used to study the allocation of photoassimilates and trade-offs between carbon sources and sinks (Rademacher et al., 2019). Girdling involves the removal of bark, phloem and cambium, resulting in an abrupt shut-down of the carbon supply of new assimilates from the canopy to organs downstream of the girdle (Maunoury-Danger et al., 2010). Girdling also induces the accumulation of non-structural carbohydrates (NSCs) above the girdle (Jordan & Habib, 1996), with this response weakening with the distance from the girdle (Goren et al., 2010). Girdling does not uniquely alter plant carbon function, as it also dries out the outer xylem rings, leading to a loss of hydraulic conductance and overall tree transpiration (De Schepper et al., 2010; Domec & Pruyn, 2008). This explains anatomical changes observed in conifers, including suppressed xylem production, enhanced phloem development, enlarged axial parenchyma cells and reduced size of conducting elements (Serkova et al., 2024). The post-girdling increase in cell-wall thickness may be due to an increase in NSCs by accumulation of photosynthates above the girdle (Jordan & Habib, 1996; Winkler & Oberhuber, 2017).

Here, we combine the long-term (2000–2022) assessment of IADF formation and its climate drivers with a short-term (one growing season) girdling experiment carried out on branches of Spanish juniper (*J. thurifera* L.) individuals inhabiting a semi-arid site. To monitor the impacts of girdling on carbon availability and radial growth dynamics, we analyzed NSC concentrations in sapwood and leaves and used automatic point dendrometers, respectively. Our aims were: (i) to monitor the impacts of girdling on branch diameter and NSC concentrations; and (ii) to test whether girdling triggered bimodality and latewood IADF production. We hypothesized that latewood IADF production depends on sufficient late-summer moisture availability (cf. Camarero et al., 2023). By focusing on seasonally dry forests and a tree species showing facultative bimodal growth and IADF formation, our general purpose is to contribute knowledge to disentangle the roles played by carbon sinks and sources on radial growth.

## 2. Materials and methods

### 2.1. Study site, tree species and climate data

We performed the experiment in the 'Vedado de Peñaflor' (41.7834° N, 0.7217° W, 374 m a.s.l.), a well-preserved mixed pine-juniper forest subjected to semi-arid climate conditions and located near Zaragoza city (north-eastern Spain). The dominant tree species is Aleppo pine (*Pinus halepensis* Mill.), with a mean basal area of 8.0 $m^2$ $ha^{-1}$, whereas arboreal (Spanish juniper, *J. thurifera* L.) or shrubby junipers (Phoenicean juniper, *Juniperus*

*phoenicea* L.) are also frequent. Other shrub and sub-shrub species include: *Pistacia lentiscus* L., *Rhamnus lycioides* L., *Salvia rosmarinus* Spenn., *Thymus* spp., *Genista scorpius* (L.) DC., *Cistus clusii* Dunal, *Globularia alypum* L. and *Ephedra nebrodensis* Tineo.

According to data from the nearby Zaragoza-airport station (41.6600° N, 1.0044° W, 249 m a.s.l., period 1941–2024), the mean annual temperature and total annual precipitation are 14.8°C and 358 mm, respectively. The period with negative water balance goes from May to October, with a mean soil moisture content (measured at 10–15 cm depth, period 2007–2011) of 10.8% (Camarero et al., 2010, 2021). Soil moisture content peaks from February to May (30–35%) and reaches its minimum values from July to September (5–10%). Soils are basic, with gypsum and marl forming the parent rock material. No anthropogenic disturbance (wildfire, thinning or logging) has affected the study site at least since the 1970s. The year 2022 was extremely dry and warm, with a total annual precipitation of only 222 mm (Supplementary Figure S1). In that year, winter precipitation, which is critical for tree radial growth in the study area (cf. Pasho et al., 2012), was only 30% (20 mm) of the long-term average (68 mm).

Monthly climate data were obtained from the 4-km Terraclimate dataset (Abatzoglou et al., 2018), which provides homogeneous, checked series for several variables such as mean maximum temperature ($T_{max}$), mean minimum temperature ($T_{min}$), precipitation (Prec.), soil moisture (SM), climate water deficit (CWD, as difference between precipitation and potential evapotranspiration) and vapor pressure deficit (VPD). Monthly reference evapotranspiration was calculated using the Penman Montieth equation (Allen et al., 1998). These data were highly representative of the study location and showed good correlations with local climate series. For instance, the series of August $T_{max}$ from Terraclimate and Zaragoza airport were highly correlated ($r = 0.934$, $p < 0.0001$, period 2000–2022). Local climate data recorded every hour (air temperature, precipitation) were obtained from an automated weather station (height = 1.5 m) situated in an open area in the study site (Campbell CR-1000, Campbell Sci., Logan, USA).

To assess drought severity, we obtained weekly and monthly values of the Standardized Precipitation-Evapotranspiration Index (SPEI), which accounts for the cumulative effects of climatic water balance. We used the 1-$km^2$ Spanish SPEI dataset (period 1960-2023) available at the webpage https://monitordesequia.csic.es/ (Vicente-Serrano et al., 2017). The summer of 2022 was exceptionally dry since 1960, as indicated by the very low SPEI values (Supplementary Figure S1). The CWD values of the hydrological year (October 1 to September 30) were −843 mm in 2020–2021 and −1031 mm in 2021–2022, whereas the 2020–2025 average was −715 mm.

### 2.2. Experimental design

We selected five dominant, mature and apparently healthy *J. thurifera* individuals growing at the study site and showing similar diameters at 1.3 m (14–22 cm) and canopy height (5.1–5.5 m). On 12 June 2021, after the spring growth peak, we selected two branches located at mid canopy with a diameter ranging from 7 to 15 cm in each individual (Figure 1a). Branches were selected for girdling because we assumed they were autonomous units for carbon transfer and storage, and because the juniper species is protected in the study area, which precluded girdling the main stem due to protection measures. Branches corresponded to girdled and non-girdled (control) branches. One branch per individual was girdled, and the other remained as a control for a total of 10 branches

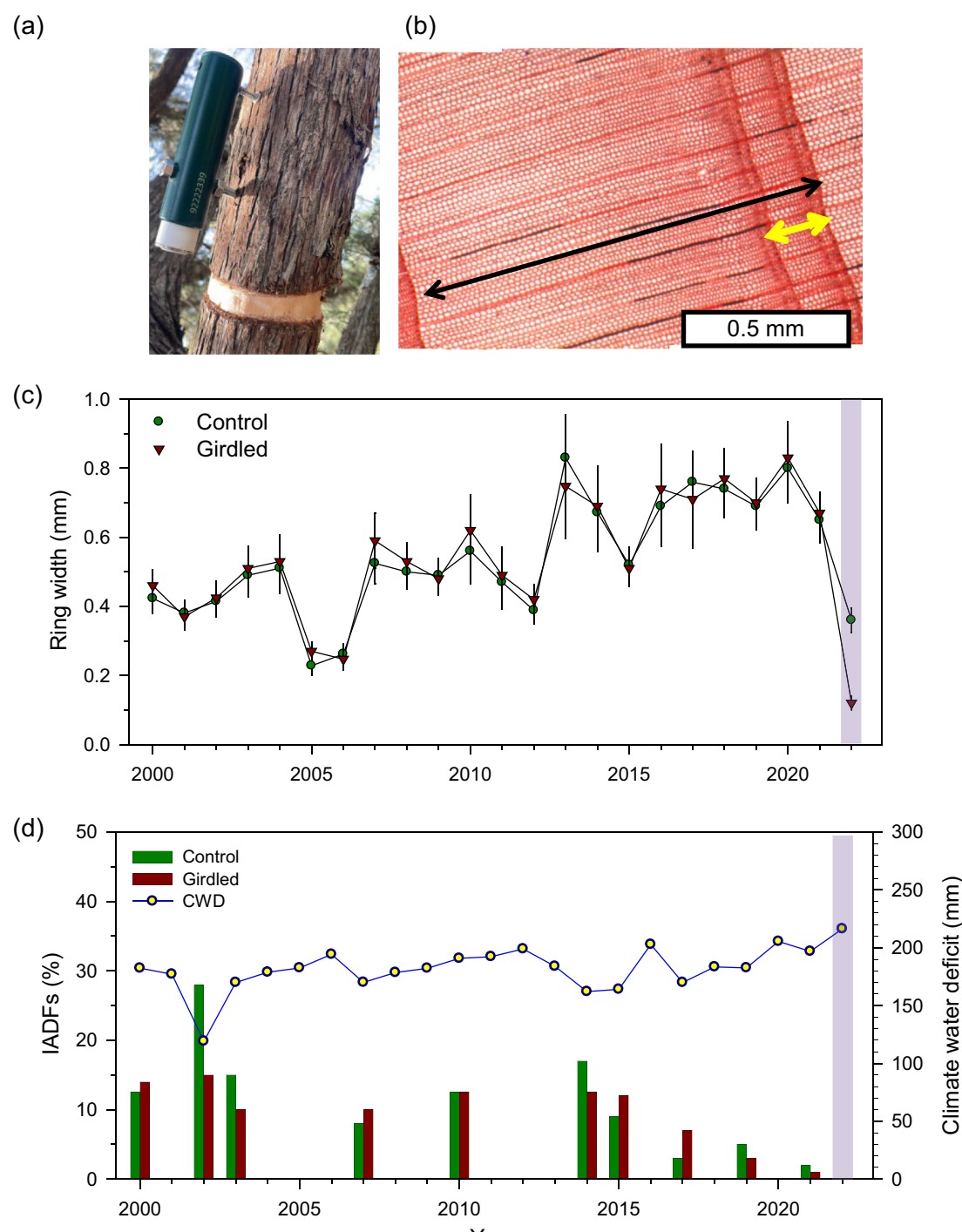

**Figure 1.** Views of (a) girdled branch with a point dendrometer attached in *J. thurifera*, (b) latewood IADF (yellow arrow) observed in the 2002 tree ring (black arrow), (c) ring-width data (means ± SD), and (d) frequency of IADF production in control (green bars) and girdled branches (brown bars; the boxes indicates the post-girdling year, 2022). The blue line shows the August climate water deficit (right *y* axis).

(5∗2). Branches were collected in early December 2022, i.e. 17 months after girdling, to analyze IADFs and the concentration of NSCs during 2022.

### 2.3. Dendrometer records and processing

Automatic point dendrometers (TOMST, Prague, Czech Republic) were used to monitor changes in branch diameter ('branch ΔD' hereafter) at a high temporal (15 min) and spatial (0.27 μm)

resolution from April until November 2022. Onset of radial growth of *J. thurifera* occurs from mid to late April in the study area, according to xylogenesis data and dendrometer records (Camarero et al., 2010, 2021). Dendrometers additionally registered air temperature, which was used in subsequent analyses. They were placed above the girdle and on the living phloem after carefully removing the dead bark to avoid hygroscopic effects related to bark presence. The sensor tip was placed at 11–12 cm from the girdle (Figure 1a). In addition, we placed band dendrometers (EMS DRS26, Brno,

Czech Republic) at 1.3 m in the stems of five junipers to record changes in stem diameter with hourly resolution during a wet year (2020) and to compare them with those obtained for the 2022 dry year. Means were obtained for the period from January to October, encompassing the whole growing season.

Dendrometer data were processed with the treenetproc package (Haeni et al., 2020; Knüsel et al., 2021) in R software (version 4.5.0; R Development Core Team, 2025). The package is based on the zero growth concept, which separates growth and water-related components of the dendrometer series assuming that growth begins when the previous maximum diameter is exceeded and ends when shrinkage occurs (Zweifel et al., 2006, 2016). Following a visual quality check, data pre-processing included: (i) excluding one dendrometer time series from controls due to persistent branch shrinkage and loss of contact with the stem surface, resulting in invalid zero readings, (ii) zeroing all time series to May 17 due to a dendrometer malfunction until this date. No further corrections were necessary. From the sub-hourly time series of branch $\Delta D$, two daily transpiration-driven diameter ($D$) extremes were detected: daily $D$ maxima ($D_{max}$) were recorded around 6:00 a.m., while daily $D$ minima ($D_{min}$) were recorded around 3:00 p.m. (Supplementary Figure S4). Then, we calculated the daily time series of (i) branch (nocturnal) swelling as the difference between (midday) $D_{min}$ and the following day (predawn) $D_{max}$, and (ii) the branch (diurnal) shrinkage as the difference between $D_{max}$ and $D_{min}$ within a day.

### 2.4. Ring-width data, IADF records and wood density

After sampling, branch cross-sections were air-dried, sanded with different sandpaper grain until rings were visible and scanned at a resolution of 2,400 dpi (Epson Expression 10,000). Two radii of each branch from opposite sides were visually cross-dated under the stereomicroscope from the bark to the pith (Fritts, 1976). The width of the annual rings was measured with 0.001 mm of resolution on scanned images using the CDendro-CooRecorder program (Maxwell & Larsson, 2021). The validation of the visual cross-dating process was undertaken by using the COFECHA software that compares the correlation between the individual series of ring-width and the mean sites series (Holmes, 1983). The IADFs frequency was obtained by counting their presence along the two cross-dated radii of each branch under the stereomicroscope. Only rings with IADF presence in both radii were considered to calculate an annual frequency (%) of IADFs for the common period 2000–2022. This was done by summing the observed IADFs and calculating relative frequencies with respect to the measured branches every year. The IADFs were formed in the mid to late wood (Figure 1b and Supplementary Figure S2). The IADF frequency was standardized and detrended by fitting linear regressions and keeping the resulting residuals.

To quantify the wood density of *J. thurifera*, we sampled 10 (length 15–20 cm, diameter 5–7 cm) branches from the upper canopy of five individuals (two branches per individual). A similar protocol was followed for the coexisting *Juniperus phoenicea* to compare both species. Wood density was estimated using the water displacement for volume and oven drying of samples at 70°C for 72 h for mass (Fajardo, 2025).

### 2.5. NSC concentrations

The concentrations of starch and soluble sugars in the branch sapwood and leaves of girdled and non-girdled branches were measured. Although phloem has higher NSC concentrations than xylem, sapwood and leaves are by far the largest tree NSC reservoir due to their greater volume and biomass (Simard et al., 2013). Wood and leaf samples (ca. 50 g) were placed in a cooler and taken immediately to the laboratory, where they were freeze-dried and milled to a fine powder in a Retsch M400 ball mill (Haan, Germany) prior to analysis. Soluble sugars (SS) were extracted with 80% (v/v) ethanol in a water bath at 60°C, and their concentration was determined colorimetrically using the phenol–sulphuric method (Buysse & Merckx, 1993). The starch and complex sugars that remained in the undissolved pellet after the ethanol extraction were enzymatically reduced to glucose using amyloglucosidase (0.5% amyloglucosidase 73.8 U/mg, Fluka 10115) and analyzed following Palacio et al. (2007). NSCs measured after ethanol extraction are referred to as soluble sugars, and those measured after enzymatic digestion are referred to as starch. The sum of soluble sugars and starch is referred to as total non-structural carbohydrates (NSC).

### 2.6. Data analyses

The R software (version 4.5.0; R Development Core Team, 2025) was used for data analyses. Mann–Whitney $U$ tests were used to compare inter-annual growth rates, IADF production and NSC concentrations between control and girdled branches, given that the percentage of IADFs did not follow a normal distribution. Spearman correlations were calculated between the annual frequency of IADFs and August climate variables based on previous analyses on xylogenesis and dendrometer data (Camarero et al., 2010, 2021, 2023).

We assessed the amplitude of daily shrinkage and swelling cycles in branches using linear mixed-effects models (*lme4* package; Bates et al., 2015). To account for the potential lack of normality, dendrometer data were log ($x + 1$) transformed. The initial model included treatment (girdling vs. control) as a fixed effect, with random intercepts for tree and branch nested within tree. As rain events appeared to influence the daily shrinkage-swelling amplitude, we subsequently fitted models that incorporated the interaction between treatment and rain, considering rain as a qualitative (rainy vs. non-rainy day) or a quantitative (rain amount) fixed effect. We assumed a normal distribution of the model residuals and homoscedasticity, which were verified using diagnostic plots (Santos Nobre & Singer, 2007). Additionally, we used generalised additive mixed models (GAMMs) to explore the shape of the relationship between rainfall amount and shrinkage-swelling amplitude (*mgcv* package; Wood, 2011). These models allowed for separate, flexible (non-linear) responses for each treatment group, along with random intercepts for tree and branch.

## 3. Results

### 3.1. Growth, IADF formation and wood density

We built a robust ring-width chronology for the common period 2000–2022. The mean ring width was $0.54 \pm 0.04$ mm (mean $\pm$ SE; control branches, $0.53 \pm 0.04$ mm; girdled branches, $0.55 \pm 0.06$ mm; $U = 242$, $p = 0.62$), and the mean correlation between ring-width series was $0.82 \pm 0.11$. Growth was strongly reduced in 2022 in girdled branches (Figure 1c), and this was confirmed by stem dendrometer records of the wet and dry 2020 and 2022 years, respectively (Supplementary Figure S3). In 2022, radial growth was significantly lower in girdled than in control branches ($0.12 \pm 0.01$ vs. $0.36 \pm 0.04$ mm, $U = 0.10$, $p = 0.01$; Figure 1c). The

**Table 1.** Spearman correlations calculated between the annual frequency of latewood IADFs and August, September and August–September climate variables ($T_{max}$, mean maximum temperature; $T_{min}$, mean minimum temperature; Prec, precipitation, Soil m., soil moisture; CWD, climate water deficit; and VPD, vapor pressure deficit).

| | Period | | |
|---|---|---|---|
| Climate variable | August | September | August–September |
| $T_{max}$ | −0.150 | −0.130 | **−0.324** |
| $T_{min}$ | −0.093 | −0.134 | **−0.276** |
| Prec | **0.442** | 0.173 | **0.348** |
| Soil m. | −0.039 | 0.087 | 0.089 |
| CWD | **−0.604** | **−0.333** | **−0.590** |
| VPD | −0.241 | −0.217 | **−0.450** |

Bold coefficients indicate significant correlations ($p < 0.05$) between IADFs and climate variables.

annual frequency of latewood IADFs did not significantly differ ($U = 260$, $p = 0.92$) between control (mean ± SE = 4.87 ± 1.57 %) and girdled branches (4.22 ± 1.20 %; Figure 1d). Considering the IADF production of all branches across years, correlations with August were higher, in absolute terms, than those with September or August-September climate conditions (Table 1). IADF frequency was negatively correlated with the August water balance deficit ($r = −0.604$, $p = 0.002$), i.e. wetter August conditions enhanced IADF production. Regarding wood density, *J. thurifera* (mean ± SD) showed significantly lower values than *J. phoenicea* (0.57 ± 0.06 vs. 0.78 ± 0.09 g cm$^{-3}$; $U = 0.01$, $p = 0.001$).

### 3.2. Stem radial responses to girdling

Throughout the period of branch $\Delta D$ monitoring in 2022, daily mean temperatures increased progressively, rising from 0°C in April to a peak of 33°C in mid-June, and remained above 20°C until mid-September (Figure 2a). Only 101 mm of rainfall was recorded from June to September (Figure 2b). None of the monitored branches showed net diameter increase, regardless of the girdling treatment (Figure 2b). Instead, we observed a progressive branch shrinkage, consistent with elastic fluctuations related to branch dehydration. Branch diameter reductions were periodically interrupted by sharp increases following rainfall events; however, these transient recoveries did not restore the initial diameter.

At the daily timescale, diurnal shrinkage tended to exceed nocturnal swelling, regardless of the girdling treatment (Figure 3), consistent with the net reduction in branch $\Delta D$ observed over the monitored period. The amplitude of these daily cycles typically remained below 30 μm day$^{-1}$. However, rainfall events triggered substantial increases in their amplitude, with daily fluctuations reaching up to 140 μm day$^{-1}$. Across the entire experimental period, treatment (girdling vs. control) did not significantly affect shrinkage ($p = 0.34$) or swelling ($p = 0.25$) amplitude. However, models incorporating rainfall revealed significant effects. Both shrinkage and swelling amplitudes were higher on rainy days compared to non-rainy days ($p < 0.0001$). Furthermore, the amplitude of both swelling and shrinkage increased significantly with the amount of rainfall ($p < 0.0001$). This rain effect was more pronounced in girdled branches than in control branches, as indicated by a significant treatment:rain interaction ($p < 0.0001$; Figure 4). The GAMMs confirmed significant effects of rainfall on both shrinkage and swelling (all smooth terms $p < 0.0001$). Visualization of the treatment effect (girdled–control) across the rainfall gradient revealed a

consistently positive influence of girdling on both processes above ca. 5 mm of rain (Supplementary Figure S5). The pattern of this effect was non-linear (parabolic) for swelling, whereas it was more linear for shrinkage.

### 3.3. NSC concentrations in sapwood and leaves

NSC concentrations were higher in leaves (ca. 12 %) than in sapwood (ca. 2 %) (Figure 5). In leaves, we found significantly ($p < 0.05$) higher starch ($U = 1$, $p = 0.017$) and NSC ($U = 3$, $p = 0.037$) concentrations in girdled than in control branches. In branch sapwood, we found no differences in soluble sugar ($U = 4$, $p = 0.096$) and NSC concentrations ($U = 8$, $p = 0.360$) between girdled and control branches.

### 4. Discussion

Girdled and control juniper branches showed permanent and progressive shrinkage after installing the dendrometers in spring (Figure 2). Thus, according to the zero growth concept (Zweifel et al., 2016), no radial growth occurred. Following this theoretical framework, branch diameter variations could be attributed to elastic shrinkage-swelling processes due to wood water depletion or replenishment, respectively. Nevertheless, we observed the formation of earlywood in 2022, which reflects limited growth during periods of branch water depletion and shrinkage, but also challenges the zero concept framework. Dendrometers did not record a net stem growth, regardless of girdling, but branches showed continuous shrinkage (Figure 2). This finding suggests that new xylem cells can be produced, even under dry conditions, and these growth processes may not be accurately recorded by point dendrometers, which only monitor a particular place of the stem and may be more sensitive to shrinking/swelling dynamics. We acknowledged the limitation of our data since the growth of the study species starts in April and the dendrometers were zeroed in mid-May. To address this issue, we plan to compare xylogenesis and dendrometer data in the future. The long-term evaluation of latewood IADF production pointed to the importance of late-summer (August) drought as previously reported (Camarero et al., 2010, 2021, 2023; Salomón & Camarero, 2025). The 2022 exceptionally dry conditions probably hampered IADF production regardless of girdling (Figure 1).

Girdling negatively impacted xylem growth above the girdled zone by blocking phloem transport and reducing cambium dynamics and xylem water flow, at least close to the girdle, as has been previously reported (Serkova et al., 2022, 2024; Tombesi et al., 2014). Starch and NSC concentrations increased in leaves above the girdling, likely due to phloem blockage and accumulation of photosynthates above the girdled part of the branch (Rademacher et al., 2019). Importantly, girdled branches were more responsive to rain pulses and showed a higher elastic capacity to rehydrate (and subsequently dehydrate) than control branches (Figure 4), with this effect saturating upon a certain rainfall amount threshold (Supplementary Figure S5). The enhanced amplitude of branch $\Delta D$ to rain pulses in girdled branches could not be attributed to higher concentrations of osmolytes (Blumstein et al., 2023), because NSC content in xylem was not significantly different among controls and girdled branches (Figure 5). We speculate that abundant leaf starch reserves of gridled branches may be hydrolyzed into soluble sugars (Richardson et al., 2013), which are then transported downstream via the phloem, accumulating above the girdled area. Their function as osmolytes could subsequently explain the more

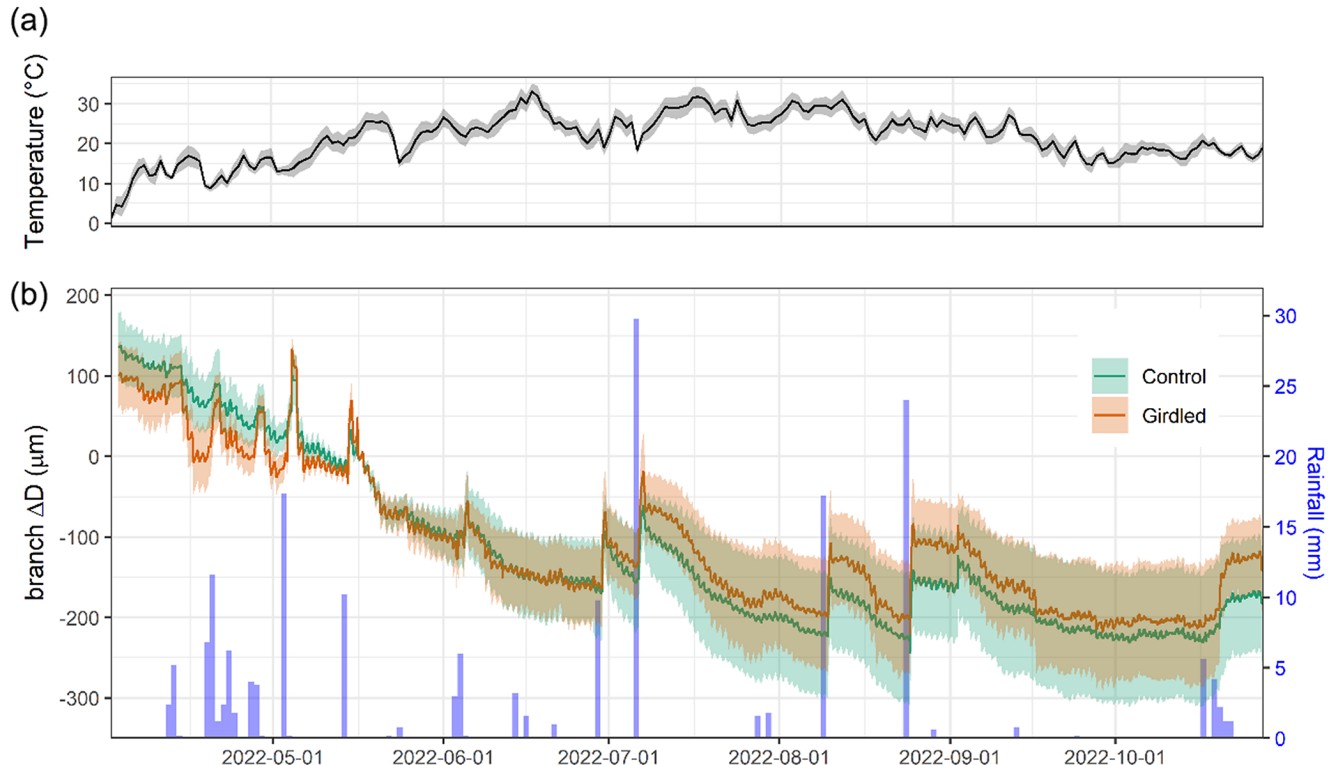

**Figure 2.** (a) Mean daily temperature and (b) variations in diameter measured in girdled (brown line) and control (green line) *J. thurifera* branches (means ± SEs) and daily rainfall records (blue bars, right y-axis) over the experimental period (from April to November 2022).

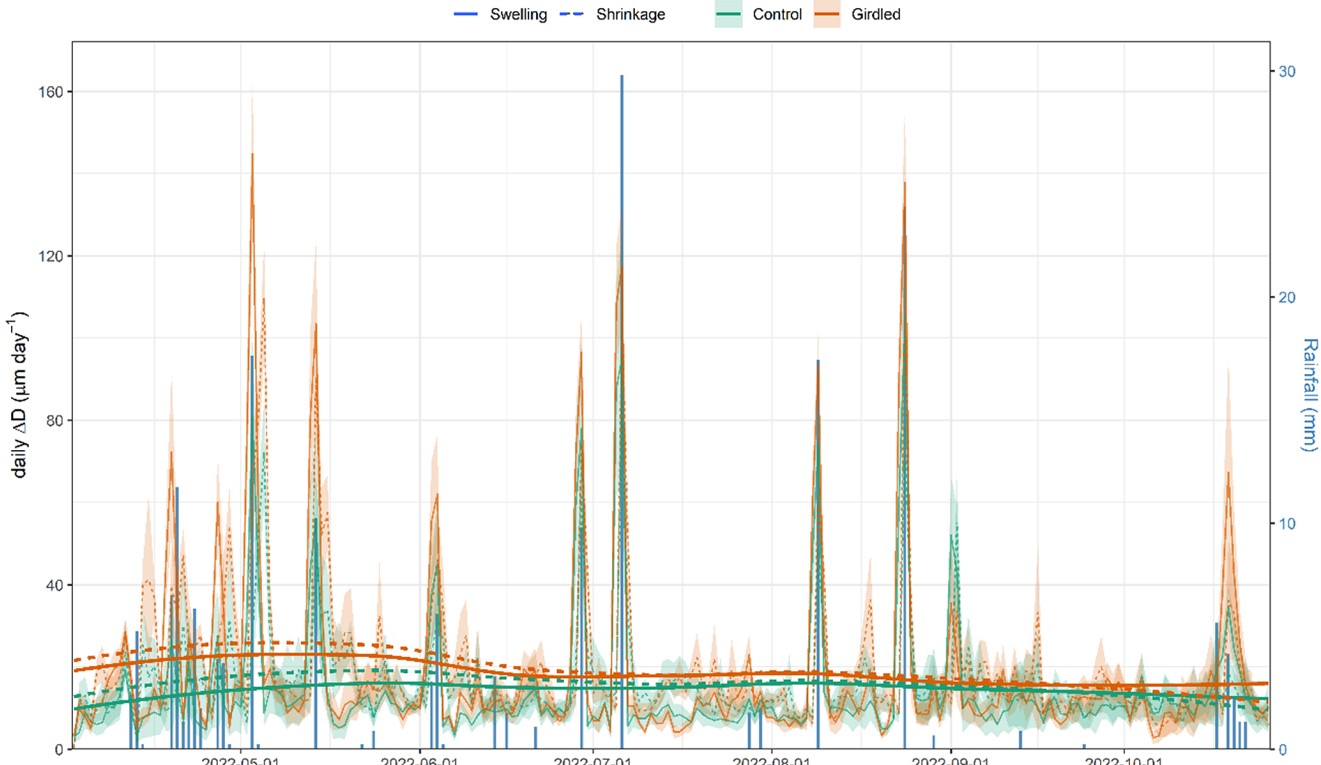

**Figure 3.** Daily patterns of branch diameter swelling and shrinkage (continuous and dashed lines) in girdled and control (brown and green colour) *J. thurifera* branches. *Note*: Lines and shaded ribbons show mean ± SE. Long-term trends were highlighted using locally estimated scatterplot smoothing (loess). Blue bars show rainfall values (right y-axis).

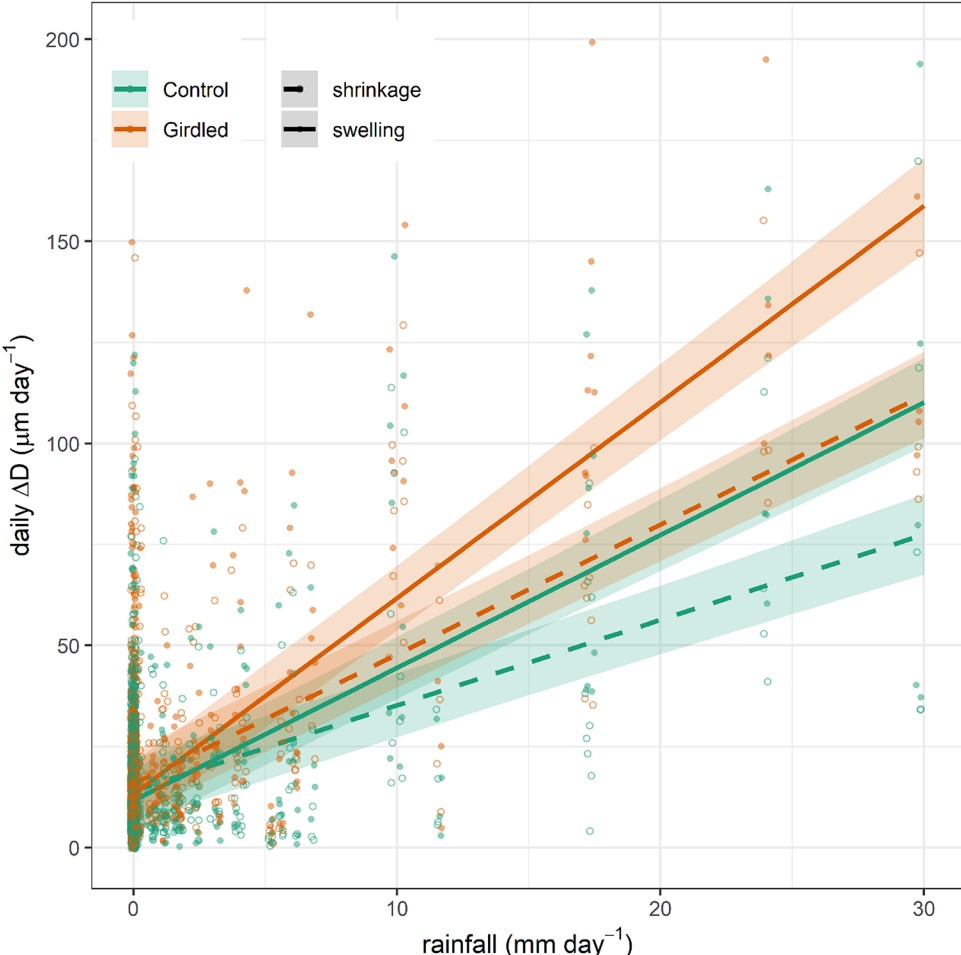

**Figure 4.** Linear relationships between rainfall amount and daily diameter swelling and shrinkage (continuous and dashed lines) in girdled and control (brown or green colour) branches of *J. thurifera*. Lines and shaded ribbons show means ± SE.

elastic rehydration response of girdled branches to rain pulses, as recorded by the dendrometers. Following these NSC allocation patterns, the observed ΔD signal is more likely attributable to swelling and shrinking in the living tissues beyond the cambium (phloem and bark) than those in the sapwood xylem. Notably, junipers showed rapid stem shrinkage after spring rainfall events under mild conditions and also after heavy rainfall events under warmer conditions in summer (Figure 2). These rapid responses could be explained by the hygroscopic nature of the low-density wood of *J. thurifera*, and its anisohydric ('water spender') behavior. However, the high variability in stem swelling/shrinkage dynamics between periods and sites depends on additional traits, in addition to wood density and stomatal closure, including water storage tissue size, elasticity and capacitance (Dietrich et al., 2018; Salomón et al., 2017), but the underlying mechanisms remain poorly understood. In general, stem or branch shrinkage amplitude increases under dry conditions (elevated evaporative demand, limited soil water supply), but with sufficient internal water storage to allow water release from elastic tissues contributing to transpiration (Steppe et al., 2006; Zweifel et al., 2001). Further research is needed to link drivers such as the compressibility of the elastic storage tissues with functional or physiological traits such as wood density or stomatal conductance rates.

Girdling did not trigger the formation of latewood IADFs in 2022, a pattern which may be attributed to the dry conditions of

the year. Inter-annual correlations between climate variables and IADF production indicate that the main trigger of IADF formation was water availability during summer (Figure 1, Table 1). This agrees with previous research on anisohydric species (cf. Voelker et al., 2018), including Mediterranean trees such as Aleppo pine (Campelo et al., 2021; De Micco et al., 2016; Pacheco et al., 2016, 2018). Therefore, enhanced water availability in late summer and autumn can increase the turgor of cambial derivatives and their radial enlargement rate, leading to the formation of tracheids with a wide lumen within the latewood (Collado et al., 2018).

The extent to which this climatic signal on IADFs affects the carbon source and sink is still unknown. Previous research showed that girdling induced bimodal growth in Scots pine at a drought-prone Alpine valley, suggesting that belowground organs influence the carbon sink strength and hormonal signalling, albeit wetter climate conditions in this Alpine valley might have also played a role in enabling a second growth peak (Oberhuber et al., 2017; 2021). Our findings do not fully concur with this idea and reinforce the importance of climate stressors as drivers of radial growth, principally the impacts of water availability on major carbon sinks such as meristems (Körner, 2015), particularly relevant on seasonally dry Mediterranean regions. At the study site, soil drought was the main constraint of stem radial growth according to previous studies comparing manual and automatic band dendrometer data (see Camarero et al., 2021; Salomón & Camarero, 2025). Moreover, in

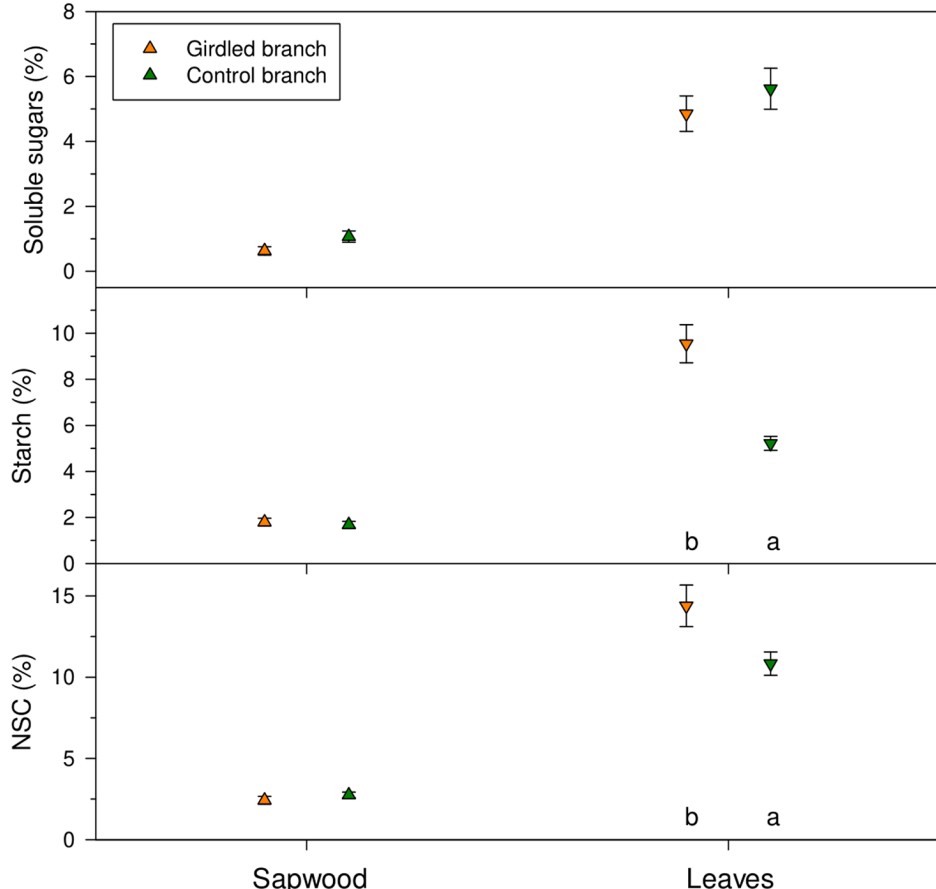

**Figure 5.** Concentrations of non-structural carbohydrates measured in sapwood and leaves of girdled (brown symbols) and control (green symbols) *J. thurifera* branches. Values are means ± SE. Different letters indicate significant ($p < 0.05$) differences between treatments according to Mann–Whitney tests.

valley bottoms of semi-arid sites, where soil moisture and nutrient availability increase, the Spanish juniper grows more, forms more IADFs and reaches larger sizes, emphasizing the importance of water availability for anisohydric species again (Camarero et al., 2023).

To conclude, we assessed the climate drivers of latewood IADF formation and performed a girdling experiment in a facultative bimodal species from a semi-arid site. No IADF formation was found in girdled branches, and late-summer water availability appears as the main trigger of IADF formation. Precluded by the extreme dry summer following treatment application, girdling did not induce bimodal growth or IADF formation. Girdled branches were more elastic in response to rain pulses and presented higher starch concentrations in their leaves. The dependence of radial growth on carbon sinks (cambium) is supported by the presented long-term climate-IADF relationships.

**Open peer review.** To view the open peer review materials for this article, please visit http://doi.org/10.1017/qpb.2026.10037.

## Acknowledgements

The authors thank E. Lahoz for performing the NSC analyses and J. Revilla for the maintenance of meteorological sensors and dendrometers.

**Competing interest.** The authors declare no competing interests.

**Data availability statement.** Data associated with this paper will be available on request from the corresponding author.

**Author contributions.** J.J.C., and A.G. conceived and designed the study; J.J.C., A.G., C.V., E.T. and M.C. conducted data gathering; J.J.C. and R.L.S. analyzed the data; J.J.C. led the writing of the manuscript; A.G., R.L.S., C.V., E.T., A.R.C. and M.C. contributed critically to the drafts. All authors have final approval for publication.

**Funding statement.** This work was funded by the Spanish Ministry of Science and Innovation (projects PID2021-123675OB-C43 and TED2021-129770B-C21).

**Supplementary material.** The supplementary material for this article can be found at http://doi.org/10.1017/qpb.2026.10037.

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
