## [Reviewer Report]

General:

In this manuscript authors focused on determining influence of girdling on radial growth of Juniperus thurifera at a semi-arid site. Although this study has some relevance in terms of the influence of environmental factors and girdling on intra-annual wood formation, methodological shortcomings exist specifically concerning (1) dendrometer records, i.e., only a 1 yr study was performed in a climatically extreme year showing only stem shrinkage, and (2) missing determination of carbohydrates (NSC) in the phloem, where NSC content is to be expected to accumulate above the girdle. Without dendrometer records in a climatically “average” year and NSC data from the phloem, it is not possible to unequivocally analyze the aims of this study.

I also miss ring width data and cross-sections of branches at the point of the dendrometer sensor for the study year 2022, when girdling occurred. Because dendrometers only recorded shrinking, all branches (controls and girdled branches) should be analyzed with respect to growth in this year, in order to depict the real radial growth and the impact of girdling. According to Fig. 1a, it looks like that the sensor tip of the point dendrometer was placed on the bark, but not on the living phloem as is usually recommended. Hence, if mounted in this way, hygroscopic influences of the bark on the dendrometer records cannot be ruled out. Furthermore, there are several inconsistencies throughout the manuscript (see specific comments below), that need to be clarified.

Specific comments and suggestions:

Lines 101ff: the second aim is unclear, because bimodal growth and IADFS can occur without girdling

Line 140: Please give more information regarding location of the weather station, e.g., height, located in an open area or below canopy?

Line 151: add canopy height of selected trees

Line 152: ?June 2021 or 2022

Line 152: “…after the spring growth peak..”.. but in line 160f it is stated that “.. dendrometers were used to monitor….from April until November…”. If there is a spring peak in radial growth, than starting dendrometer records in April might have missed some growth. Please indicate when onset of radial growth of Juniper or other tree species is expected to occur within the study area.

I also wonder why branches were selected for girdling and not the main stem at breast height. Please give reason for this.

Line 163: Dendrometers recorded temperature – it is not clear if these records were used in the analyses. Also the mean distance between the sensor tip and the girdle should be mentioned. Was the sensor tip placed on the bark (see Fig. 1b) or on the living phloem? The latter is recommended to prevent influence of bark hygroscopicity on dendrometer traces?

Line 170: one dendrometer was excluded – from controls or girdled tree?

Line 175: change specified time to a.m. and p.m.

Line 183: Were selected radii from opposite sides or in a specific angle, e.g., 90 °?

Line 199ff: Any reason why NSC content was not determined in the phloem?

Line 235: Please show ring width chronologies for the period 2000-2022 for both, controls and trees girdled in 2022. Also ring width and tracheid dimensions in 2022 (as in Fig. S2) should be compared between controls and girdled trees. If radial growth occurred in 2022 and was not detected in dendrometer records, this must be thoroughly clarified.

Line 250: “volumetric growth”: unclear meaning, please reword

Line 262f: repetition of sentence starting with “However, models incorporating….”

Line 284: “a second peak of growth during late summer… did not occur”. In dendrometer records (Fig. 2) I see no growth at all, i.e., also spring growth is missing there.

Line 290: “…main drivers of bimodality..”. In Fig. 1c there is no IADF given for the year 2022. Please clarify.

Line 301: NSC content in xylem was not significantly different among controls and girdled branches and phloem NSC was not determined. Hence, the given interpretation of “the enhanced amplitude of branch ΔD to rain pulses” is mere speculation.

Line 316: There are no data or graphs which show radial growth in 2022 in controls and girdled branches. Please add.

Line 317ff: The finding that water availability in late summer after a drought period induces IADFs is well known. The girdling effect on radial growth is not well documented, because dendrometer records show no growth increment and wood formation in 2022 is not shown, i.e., no cross sections of tree ring formation in 2022 are depicted.

Line 347f: “In the long-term (2000-2022), no differences in IADF formation were found between girdled and control branches,…”. Rewording necessary, because girdling occurred only in the year 2022.

Fig. 1a: It looks like that the dendromter sensor is placed on the bark, not the living phloem, which however is recommended to prevent influence of shrinking and swelling of the bark (hygroscopic tissue) on dendrometer records.

Fig. 1b: The arrows need to be placed in the correct position.

Fig. 1c: It is confusing to name branches girdled in 2022 as “girdled”. I also suggest moving this graph to supplementary material and instead show cross sections of radial growth in 2022 of a control and a girdled branch. Furthermore, it appears that the climate water deficit in August has no significant influence on IADFs. Although the August CWD has increased, there has been a decreasing trend in IADFs since 2014. Please clarify.

Fig. 2: Dendrometer records show no growth at all, but a shrinkage by about 150-200 µm during April through November. There is even shrinkage in late April through early May when several days with precipitation were recorded and daily mean temperature was below 20 °C. Such rapid shrinkage after rainfall events, which is also seen after heavy rainfall events, e.g., in early July, requires a thorough explanation. A second study year of “average” climate conditions is necessary to prove the findings.

---

## [Reviewer Report]

This manuscript Girdling increases branch capacity to rehydrate in Juniperus thurifera examines phenomena such as bimodal cambial growth and intra-annual fluctuations in wood density (IADFs). The authors intended to test the hypothesis that bimodality and IADFs may be driven by climatic factors and increased carbon levels in the cambial zone due to girdling. Studying the causes and mechanisms underlying these two phenomena is relevant for the Mediterranean region, where bimodality and IADFs are found in a number of coniferous species. However, I recommend rejecting publication of this manuscript due to numerous concerns regarding the study design and presentation of the data.

Key points:

1. In the Introduction and Abstract, the authors state that the study is related to identifying the role of carbon sinks and sources in regulating radial growth, i.e., they are declaring a very broad problem. However, the scientific question is not clearly formulated, and a review of the manuscript reveals that the study is very localized, as it was conducted on a single species and using a very limited number of methods. The scientific problem and research question should be clearly formulated, and the scope of the study should be commensurate with them.

2. The study’s novelty is lacking. Trunk girdling is a very old method, used to identify relationships between carbon distribution in trees and wood formation for over 50 years. The accumulation of NSC in trunk tissues above the girdle and in leaves, as well as changes in trunk/branch diameter above the girdle, have been repeatedly demonstrated in the work of other researchers.

3. The choice of objects and methods is very poorly justified, suggesting that the study design was insufficiently planed in advance. I have fundamental questions about literally every part of the Materials and Methods section:

• Why were trees growing in natural conditions chosen as objects, rather than experimental tree plantings with controlled irrigation?

• If 2022 was atypical in terms of weather and did not allow for the bimodal growth phenomenon to be studied, then why wasn’t the experiment repeated in subsequent years?

• Why did the authors compare the frequency of IADFs in girdled and control branches for the period 2000-2022, if girdling was only performed in June 2021?

• Why were correlations between the frequency of IADFs and climate parameters only determined for August?

• Why were dendrometers only installed in 2022, even though girdling was performed in 2021?

• Why was the NSC concentration in the sapwood of branches determined, but not in the bark/phloem?

• Why was only NSC determined and not the content of other osmotically active substances or the osmolality of phloem and xylem exudates?

• Why were the water potentials of leaves/shoots not measured in the pre-dawn and midday hours on girdled and control branches?

The Materials and Methods section should be written with the utmost care, with all key aspects of the study well-justified and described. The authors used the girdling method, for which it is well known that the distance from the girdle and the duration of the period following girdling significantly influence the dynamics of NSC content and the structure of conductive tissues. Therefore, special attention should be paid to these aspects when describing the methods: what, where, and when was analyzed, and why.

4. I consider it unnecessary to comment on the Results section, since the results can only be assessed if there are no questions about the methodological part of the work.

5. The Discussion section is written in a highly speculative manner. In some cases, the authors discuss data not presented in the Results section, for example, page 13, lines 291-293:

Nevertheless, we observed the formation of a thin earlywood in 2022, which reflects limited growth during periods of branch water depletion and shrinkage.

The discussion should clearly align with the results obtained and support the authors' conclusions.

6. Figure 1b and the graphical abstract on page 31 need significant improvement.

---

## [Editor Report]

Your manuscript has been fully evaluated by two independent reviewers. While they both acknowledge the relevance of your study, they also point several methodological shortcomings, and a lack of justification and clarity in the methods and experimental design. They are asking for more data, or at least more consistency between your stated research questions and the results actually presented.

Addressing these points requires substantially revising your manuscript. I thank you in advance for the effort, and I look forward to receiving the revised version.

---

## [Reviewer Report]

In this manuscript, Camarero and co-authors use the interaction of drought and girdling to study tree carbon and water relations and the occurrence of intra-annual density fluctuations. To do so, they set up a small experimental unit of 5 Juniper trees on which they monitor two branches: one girdled and one control. Dendrochronological analysis of branches collected at the end of one year of treatment (2022) show that intra-annual density fluctuations are associated with wetter-than-average year. There was no IADF formation in 2022, which was much drier than average. Contrary to previous study, they did not find that girdling promoted IADF occurrence, but this result is likely due to 2022 being among the driest years of the considered period. Yet, they did find higher starch concentration in leaves of girdled branches (not sapwood), compared to control, as well as larger amplitude of swelling and shrinking dynamics, suggesting a role of osmolytes. Despite small population size and somewhat rudimentary analyses, the results from Camarero et al support their conclusion that suggest an interaction of water and carbon status in driving growth, though water status seems to have had a decisive role here.

The manuscript is clearly written and the figures are easy to follow. However, several points need to be addressed.

A key result, that growth was strongly reduced in 2022 in girdled branches, is not mentioned in the text. The implications of the negative effects of girdling above the girdling zone should be discussed. In addition, the dendrometer measurements of stem growth are not described or discussed.

The statistical analyses require clarification and, in some cases, improvement.

• The correlation analysis with IADF is reduced to one sentence. Why was August precipitation selected rather than other climate variables or time periods? Was a moving-window analysis performed?

• The mixed-effects modelling needs more explanation: What distribution was assumed for the residuals, and was this assumption checked?

• Why was the Mann–Whitney test used?

• How was CWD calculated? More information is needed to place the 2022 drought in context relative to other years, not only through CWD.

The abstract figure is not self-explanatory. Please consider substantial redesigning.

Line 278: Please report the exact p-value.

Lines 329–332: Results on wood density and on the second Juniperus species are mentioned for the first time in the Discussion; these should be introduced earlier in the Results.

---

## [Editor Report]

Your revised manuscript has been fully evaluated by an independent reviewer, different from the two previous ones. While he/she acknowledges that your main conclusion is well-supported by your data and analyses, she/he thinks that important points should be clarified and/or improved.

Addressing these points requires a new revision of your manuscript. I thank you in advance for the extra effort, and I look forward to receiving the revised version.

---

## [Reviewer Report]

Regarding the mixed-effects models, the authors now specify that a normal distribution was assumed. However, this assumption is questionable for dendrometer data, which are often non-normally distributed due to skewness and temporal autocorrelation (as can clearly be seen in Fig 3 and 4). Did the authors examine the distribution of model residuals to verify this assumption? If not, this should be done and reported. Alternative distributions (e.g. gamma) or appropriate transformations should be considered if normality is violated.

The Materials and Methods mention a comparison between band and point dendrometers, but no such comparison is presented in the Results. This comparison should be explicitly shown, or the corresponding statement should be removed from the Methods to avoid confusion.

The relevance of the wood density measurements is unclear. While the authors aim to argue that rapid stem shrinkage in J. oxycedrus is related to elastic wood properties and anisohydric behaviour, differences in wood density with J. phoeniceae alone are insufficient to back this claim. The authors should either substantially strengthen this section with additional references and mechanistic justification, or consider removing this part of the Discussion.

Finally, dendrometers were potentially installed or zeroed (17 May) after the onset of cambial growth, which the authors mention is expected to occur in April for this species in their study area. If this is the case, the conclusion that growth occurred during periods of stem shrinkage is not fully supported by the data. This limitation should be clearly acknowledged as a caveat in the Discussion. Alternatively, the authors should remove or substantially revise this section to avoid overinterpretation.

---

## [Editor Report]

Your revised manuscript has been fully evaluated by the reviewer who already reviewed the previous version. While you took into account his/her previous comments, a few points still need improvement. Please highlight clearly in your revised manuscript which parts of the text have been amended.